# Development of Protoplast-Based Gene Editing System for Areca Palm

**DOI:** 10.3390/plants14060832

**Published:** 2025-03-07

**Authors:** Hao Nie, Saira Batool, Yin Min Htwe, Xiaomeng Fang, Dapeng Zhang, Peng Shi, Zhiying Li, Mingjun Ma, Hanlu Su, Qun Yu, Xiangman He, Yong Wang

**Affiliations:** 1National Key Laboratory for Tropical Crop Breeding/Coconut Research Institute/Sanya Research Institute, Chinese Academy of Tropical Agricultural Sciences, Sanya 572025, China; niehao344@163.com (H.N.); sairabatool0799@outlook.com (S.B.); yinminhtwemgk@gmail.com (Y.M.H.); zhangdp@catas.cn (D.Z.); ship@catas.cn (P.S.); lizhiyingalien@gmail.com (Z.L.); mmjyxh@163.com (M.M.); 13905483918@163.com (H.S.); yuqun1998@gmail.com (Q.Y.); 18389739413@163.com (X.H.); 2College of Horticulture and Forestry Sciences, Huazhong Agricultural University, Wuhan 430070, China; 3National Nanfan Research Institute, Chinese Academy of Agricultural Sciences, Sanya 572000, China; fangxiaomeng222@163.com

**Keywords:** areca palm, PEG-mediated transformation, protoplast, subcellular localization, CRISPR/Cas9, *AcPDS*

## Abstract

Areca palm (*Areca catechu* L.) is an economically significant crop in tropical and subtropical regions. However, an efficient transformation and gene editing system for genetic improvement has still not been established. In this study, protoplasts were isolated from juvenile leaves, followed by PEG-mediated transformation and gene editing targeting the areca palm *AcPDS* via the CRISPR/Cas9 system. High yield (9.08 × 10^6^ cells/g FW) and viability (91.57%) protoplasts were isolated successfully by digestion for 5 h in an enzyme solution. Transformation efficiency (11.85%) was obtained through PEG-mediated transformation (incubation for 30 min in the mixture containing 40% PEG-4000, 400 mM CaCl_2_, 30 µg of plasmid DNA, and 100 µL of protoplasts). Furthermore, subcellular localization was established by the cotransformation of GFP and pNLS-mCherry in the protoplasts. Moreover, the editing efficiency (2.82%) of *AcPDS* using the CRISPR/Cas9 system was detected by Hi-TOM sequencing. This study established an efficient transformation and gene editing system based on protoplasts in areca palm, which will be beneficial for gene function verification and genetic improvement in areca palm and other tropical palm crops.

## 1. Introduction

Areca palm (*Areca catechu* L.) is an economically important and ornamental perennial plant [1]. It is widely cultivated in tropical regions [2] and has extensive applications in agriculture and industry. In China, areca palm is mainly distributed in Hainan, Taiwan, and Yunnan provinces [3], with over 95% of the cultivation area located in Hainan Province [4]. The areca palm industry in Hainan is a major source of income for more than two million farmers in the eastern, central, and southern mountainous regions of the province. Despite its economic importance, breeding efforts to improve the areca palm remain limited to traditional methods. To date, no effective or stable genetic transformation system has been developed for areca palm. Although the sequencing of the areca palm genome [2,5] has facilitated gene identification research, an efficient transformation and gene editing system for genetic improvement has yet to be developed.

Protoplasts are plant cells that have had their cell walls removed [6]. Selecting appropriate materials for protoplast isolation is essential for achieving a high yield and preserving protoplast viability [7]. Isolated protoplasts serve not only as an efficient source for callus culture and somatic hybridization but also as a versatile experimental system for investigating gene expression regulation and subcellular protein localization [8]. Common methods for plant protoplast transformation include PEG-mediated transformation [9], electroporation-mediated transformation [10], and microinjection-based transformation [11]. Among these, PEG-mediated transformation is widely preferred due to its simplicity, low cost, minimal equipment requirements, and consistent results [12,13]. In woody plants, protoplast transient transformation systems have been established for species such as citrus [14], rubber [15], and oil palm [16], with a few studies also reporting success in areca palm [17].

Clustered, regularly interspaced short palindromic repeat (CRISPR)/CRISPR-associated protein 9 (Cas9)-mediated gene editing enables precise manipulation of target genes and serves as a powerful tool for precision breeding in plants [18]. Its adoption is critical to addressing challenges and unlocking the potential of genetic engineering for areca palm improvement. However, to effectively apply CRISPR/Cas9 to Areca species—particularly those recalcitrant to genetic transformation—it is essential to establish efficient protocols for identifying active sgRNA targets [19]. The gRNA sequence is crucial for ensuring both targeting specificity and cleavage efficiency [20]. The targeting efficiency of sgRNA is a key determinant of successful gene editing, influenced by factors such as sequence recognition and GC content [21,22]. Additionally, the 3′ end segment of the gRNA plays an important role in the assembly of the CRISPR-Cas9 effector complex and gene editing efficiency [23,24]. The CRISPR/Cas9 technology has been successfully implemented in various crops, including woody plants such as citrus [25], oil palm [26], and rubber [27]. However, there have been no reported practical applications of CRISPR/Cas9 in areca palm until now.

In this study, we isolated protoplasts from four sample types: the callus, white leaves, light green leaves, and dark green leaves. By evaluating yield and viability, we identified the most suitable materials for protoplast isolation. Additionally, we optimized protocols for PEG-mediated transformation and applied CRISPR/Cas9 gene editing targeting the *AcPDS* gene. Our findings provide valuable insights into protoplast isolation, PEG-mediated transformation, and CRISPR/Cas9-mediated gene editing in areca palm, offering a promising resource for the genetic improvement of areca palm and other recalcitrant plant species.

## 2. Materials and Methods

### 2.1. Plant Materials

To determine the most suitable materials for protoplast isolation, four sample types were tested: the callus, white leaves, light green leaves, and dark green leaves. Callus tissues were included as they represent a dedifferentiated and rapidly dividing cell population, commonly used in tissue culture studies due to their totipotency and potential for somatic embryogenesis. White leaves were selected because they are newly emerged, unexpanded leaves with reduced chlorophyll content and softer cell walls. These features are hypothesized to enhance enzymatic digestion and protoplast release. Light green leaves represent a transitional developmental stage between white and dark green leaves, providing insight into the effects of partial chlorophyll development and cell wall lignification on protoplast isolation efficiency. Mature, fully expanded dark green leaves were included to evaluate the suitability of highly differentiated and lignified tissues for protoplast isolation. The callus used in this study was derived from tender inflorescences (≤10 cm) of the “Reyan No.1” variety of areca palm. Following surface sterilization with alcohol, the outer spathe was carefully removed under sterile conditions in a laminar flow hood. The inflorescences were then cut into 0.5–1 cm segments on sterile filter paper and inoculated onto a callus induction medium. These were cultured in the dark at 27 °C ± 2 °C for 3–4 months, with subculturing every two months. The resulting callus was considered suitable for protoplast isolation. Areca palm plants were cultivated at the experimental station of the Coconut Research Institute, Chinese Academy of Tropical Agricultural Sciences. For protoplast isolation, fresh, unexpanded small leaf sections were collected from 12 to 30-month-old areca palm trees. Three sections of the leaves were selected based on their distances from the growth point: 0–5 cm, 5–10 cm, and 10–15 cm, respectively.

### 2.2. Vector Construction

The green fluorescent protein (GFP)-expressing plasmid p35S-GFP was kindly provided by Dr Wenqiang Li of Northwest A&F University, China. This vector contains a GFP marker driven by a minimal 35S promoter, enabling the evaluation of protoplast transient transformation efficiency. The construction of pNLS-mNeon and pNLS-mCherry vectors, incorporating nuclear localization signal (NLS)-tagged fluorescent proteins, was performed following the protocol described by [28]. Primer details are provided in Table 1, and structural diagrams of all expression vectors are shown in Figure 1. To target the phytoene desaturase (*PDS*) gene in areca palm (*AcPDS*), homologous sequence alignment was used to identify the gene.

In plant gene editing, the *Arabidopsis thaliana* U6 (AtU6) promoter is widely used in many commercial editing vectors due to its broad applicability across species. It remains unclear whether the AtU6 promoter can be effectively applied in areca palm; therefore, it was preferentially selected to drive sgRNA expression in this experiment. Initially, we designed a single-target editing vector pAtU6-ZmCas9. To improve editing efficiency, we then designed a dual-target vector pAtU6-2gR-ZmCas9 incorporating two distinct sgRNAs. The 20-base pair single guide RNAs (sgRNAs) targeting exon 1 were designed using the CRISPR-P 2.0 tool (http://crispr.hzau.edu.cn/CRISPR2/) accessed on 18 August 2021, and their sequences are listed in Table 2. Two gene editing vectors were constructed by VectorBuilder (Figure 2). Plasmid DNA was extracted using the TianGen Biotech plasmid extraction kit (DP118-03, Beijing, China), and concentrations were adjusted to approximately 1000 ng/μL for protoplast transformation experiments.

### 2.3. Protoplast Isolation

The isolation of protoplasts was carried out according to the method described by Wang et al. [17], with a few modifications. The tender leaves from the unexpanded parts of areca palm seedlings were first cleaned with alcohol. The outer epidermis was peeled off to extract the white leaf tissue. The tender leaves were then cut into strips approximately 1–2 mm wide using a sharp scalpel. A protoplast separation medium was prepared containing 1.5% (*w*/*v*) cellulase R10, 0.75% (*w*/*v*) macerozyme R10, 1% (*w*/*v*) pectolyase, 0.15 mM KH_2_PO_4_, 0.7 M mannitol, 10 mM CaCl_2_, 20 mM KCl, and 20 mM MES (pH 5.7). The medium was heated in a 55 °C water bath for 10 min to inactivate proteases and ensure complete enzyme dissolution. After cooling to room temperature, 10% BSA was added to achieve a final concentration of 1% (*w*/*v*), along with β-mercaptoethanol (0.0357% *w*/*v*) and ampicillin (0.05% *w*/*v*). The mixture was inverted to homogenize and then sterilized using a 0.22 μm filter.

Approximately 20 mL of the prepared separation medium was added to a 50-mL conical tube containing the leaf strips. The tube was sealed with aluminum foil and placed in a shaker incubator, and the sample was incubated in the dark at 28 °C with gentle shaking at 60 rpm for 4–5 h. After incubation, the mixture was filtered through a 70 μm nylon mesh to separate the protoplasts. The filtered protoplast solution was gently mixed with 20 mL of W5 buffer (154 mM NaCl, 125 mM CaCl_2_, 5 mM KCl, 5 mM glucose, 1.5 mM MES, pH 5.8) and centrifuged at 150× *g* for 5 min. The supernatant was carefully discarded, the protoplast pellet was resuspended in 10 mL of W5 buffer, and the suspension was centrifuged again at 150× *g* for 5 min. Finally, the protoplast pellet was resuspended in 1 mL of W5 buffer and incubated on ice for 30 min.

### 2.4. Protoplast Yield and Viability Assay

The yield and viability of freshly isolated areca protoplasts were evaluated using the fluorescein diacetate (FDA) hydrolysis assay, as described by [29]. A 100 μL aliquot of protoplasts in W5 buffer was transferred into a 2-mL, round-bottom tube, and 1 μL of 1% FDA solution was added to achieve a final concentration of 0.01% FDA. After 5 min of incubation, the protoplasts were examined and imaged using bright-field and epifluorescence microscopy with a Zeiss Axio Imager Z2 (Zeiss, Jena, Germany) upright fluorescence microscope. The protoplast yield was assessed by diluting purified protoplast suspension and counting the cells using a hemacytometer (Watson, Tokyo, Japan) under a microscope. The yield was calculated using the formula:$$Protoplast\;yield=\frac{Total\;number\;of\;protoplasts}{Fresh\;weight\;of\;sample\;used\;(gFW)}$$

The percentage of viable protoplasts was calculated using the formula:$$Protoplast\;viabality\;(\%)=\frac{Total\;number\;of\;fluorescent\;protoplasts}{Total\;number\;of\;protoplasts}\times 100$$

### 2.5. Protoplast Transformation

Protoplast transformation was carried out using a modified polyethylene glycol (PEG)-mediated protocol for Arabidopsis [30]. Freshly isolated protoplasts were centrifuged at 100× *g* for 5 min and resuspended in MMg buffer (0.4 M mannitol, 15 mM MgCl_2_, 4 mM MES, pH 5.8) to a final density of 1.0–1.5 × 10⁶/mL. To initiate transformation, 30 µg of plasmid DNA (p35S-GFP) was added to a 2-mL, round-bottom tube, followed by 100 μL of MMg-buffered protoplast suspension. An equal volume of PEG-CaCl_2_ solution (0.2 M D-mannitol, 400 mM calcium chloride, and 40% (*w*/*v*) PEG4000) was then mixed into the tube. The transformation mixture was incubated at room temperature in the dark for 30 min. The transformation was terminated by adding 1.6 mL of W5 buffer, followed by gentle inversion of the tube. The suspension was centrifuged at 150× *g* for 3 min at room temperature, and the supernatant was discarded. The washing step with W5 buffer was repeated twice. Finally, the protoplasts were resuspended in 2 mL of W5 buffer and incubated in the dark at room temperature (28 °C) for 16 h. After incubation, the protoplasts were centrifuged at 150× *g* for 2 min, and approximately 75% of the supernatant was carefully removed. The remaining pellet was gently resuspended to minimize mechanical damage. Transformation efficiency was examined using a fluorescence microscope (Olympus, Tokyo, Japan) equipped with blue light. Green fluorescent protoplasts were counted, and the transformation efficiency (%) was calculated using the formula:$$Transformation\;efficiency\;(\%)=\frac{Total\;number\;of\;fluorescent\;protoplasts}{Total\;number\;of\;protoplasts}\times 100$$

### 2.6. Subcellular Localization

The GFP expression vector p35S-GFP, along with the nuclear localization signal (NLS) vectors pNLS-mNeon and pNLS-mCherry, were transfected using the method described in the Section 2.5, with slight modifications. To visualize the nucleus, 4’,6-diamidino-2-phenylindole (DAPI) staining was used. The concentration of DAPI staining solution for nuclear staining was increased by adding 1 µL of DAPI working solution to every 100 µL of protoplast suspension, followed by observation after incubation at room temperature for five minutes. For cotransfection, 20 μg each of the plasmids P35S-GFP and pNLS-mCherry were introduced. After 16 h of incubation in the dark, observations were performed using the confocal laser scanning microscope (LSM 900, ZEISS, Jena, Germany). P35S-GFP and pNLS-mCherry signals were visualized using the excitation/emission wavelengths of 488/525 nm and 587/625 nm, respectively. Images were captured using the ZEN 3.4 (blue edition) software.

### 2.7. Gene Editing Targeting AcPDS Gene Using the CRISPR-Cas9 System

The structure of the *AcPDS* editing vector for areca palm is illustrated in Figure 2. Transfection was performed as described in the Section 2.5, with an extended incubation period of 72 h. Genomic DNA was extracted from areca palm protoplasts subjected to PEG-mediated transformation using the TianGen Biotech plant DNA extraction kit (DP360, Beijing, China). PCR amplification was conducted using 2 × Taq Master Mix (Vazyme, Nanjing, China) with specific primers targeting the *AcPDS* gene, Ac-Hi F2 (ggagtgagtacggtgtgcACTAGGGTA GAACTGCCAGG) and Ac-Hi R2 (gagttggatgctggatggCTGAAGAGGGCTGACTTTAT). The amplified genomic region encompassed the target site. The PCR conditions consisted of an initial denaturation at 95 °C for 3 min, followed by 35 cycles of 95 °C for 15 s, 57 °C for 15 s, 72 °C for 5 s, and a final extension at 72 °C for 5 min. Amplification success was confirmed by analyzing 5 µL of PCR products through agarose gel electrophoresis. To detect mutation at the target site, the remaining 25 µL of PCR products were sequenced and analyzed using the high-throughput mutation tracking platform Hi-TOM [31].

### 2.8. Statistical Analysis

Data analysis and visualization were performed using GraphPad Prism version 8.0.2. All figures were generated using the same software. Results were presented as the mean ± standard deviation (SD) of three replicates. Statistical significance was analyzed using one-way ANOVA followed by Tukey’s multiple comparisons test to identify significant differences among groups.

## 3. Results

### 3.1. Protoplast Isolation from Areca Palm Tissues

Protoplasts were successfully isolated from the four sample types: the callus, white leaves, light green leaves, and dark green leaves (Figure 3a), by digesting them with an enzyme solution for 5 h. Among the materials, white leaves released the highest number of protoplasts (9.08  ×  10^6^ cells/g FW), significantly surpassing the yields from light green leaves (4.43 × 10⁶ cells/g FW), the callus (3.30  ×  10^6^ cells/g FW), and dark green leaves (0.78  ×  10^6^ cells/g FW) (Figure 3b,d). The viability of protoplasts derived from the three leaf sections of the areca palm plant was consistently high (Figure 3c,e), exceeding 90%: 91.57% for the white leaves, 91.50% for light green leaves, and 92.28% for dark green leaves. In contrast, protoplasts obtained from the callus exhibited significantly lower viability at 76.06% (Figure 3e). Due to their high yield and viability, protoplasts from white leaves were selected for downstream applications such as transfection and gene-editing experiments.

### 3.2. Optimization of PEG-Mediated Protoplast Transformation

To assess the suitability of isolated protoplasts for downstream transfection and gene-editing applications, we optimized a PEG-CaCl_2_-mediated transformation protocol. Freshly isolated protoplasts were transfected with the p35S-GFP plasmid. Following transfection, protoplasts were monitored continuously over 5 days to determine the optimal expression time. Green fluorescent protein (35S: GFP) expression was initially observed at 16 h post-transfection (Figure 4a) and peaked at 72 h (day 3), reaching 7.32% efficiency (Figure 4b,c). Beyond this time point, GFP expression declined. Additionally, progressive protoplast shrinkage and rupture were observed during the monitoring period.

To determine the optimal plasmid DNA concentrations for transformation efficiency, varying amounts of the p35S-GFP plasmid vector (10, 20, 30, 40, and 50 μg) were introduced into 100 μL of protoplasts (1.0–1.5 × 10^5^ cells/g FW). The resulting transformation efficiencies are shown in Figure 4b. At a DNA concentration of 10 μg, the transformation efficiency was only 0.78%. Transformation efficiency increased with higher DNA concentrations up to a certain threshold, reaching 6.48% at 30 μg. When the DNA concentration was raised to 40 μg, the transformation efficiency slightly improved to 6.78%. However, a further increase to 50 μg resulted in a significant decline in transformation efficiency to 6.09% (Figure 4b,d). There was no significant difference in transformation efficiency between 30 μg and 40 μg plasmid DNA. Considering the difficulty of plasmid enrichment, 30 μg was chosen for subsequent experiments.

This study also examined the effect of varying CaCl_2_ concentrations (100, 200, 300, 400, and 500 mM) in the PEG solution on transformation efficiency. The lowest efficiency, 2.33%, was observed at the 100 mM CaCl_2_. As the CaCl_2_ concentration increased, transformation efficiency improved, reaching 5.38% at 200 mM and 10.71% at 300 mM. The highest efficiency, 11.85%, was achieved at 400 mM. However, further increasing the CaCl_2_ concentration to 500 mM resulted in a slight decline in efficiency to 11.23%. These findings indicate that the transformation efficiency is maximized at 400 mM CaCl_2_, with a decline observed at higher concentrations (Figure 4b,e).

### 3.3. Subcellular Localization Analysis

To verify the subcellular localization of marker proteins, we transiently expressed a GFP expression vector (p35s-GFP) along with vectors containing nuclear localization signals (pNLS-mNeon and pNLS-mCherry). DAPI staining was used to visualize nuclei and the control showed no GFP signal (Figure 5a). Fluorescence from the p35s-GFP vector was observed throughout the cell membrane, cytoplasm, and nucleus (Figure 5b). In contrast, fluorescence from pNLS-mNeon and pNLS-mCherry was exclusively localized in the nucleus (Figure 5c,d), confirming successful transient transformation in the protoplasts. To further validate the correct subcellular localization of fusion proteins, we cotransformed pUN131-GFP and pNLS-mCherry. The results demonstrated strong colocalization between the markers (Figure 5e). These findings confirm that the optimized transfection procedure efficiently delivered plasmid DNA into the host cell nucleus.

### 3.4. CRISPR/Cas9-Mediated Gene Editing

Gene editing in areca protoplasts was performed using two distinct CRISPR/Cas9 vectors targeting the *PDS* gene (Figure 6a,b). Two CRISPR vectors (pAtU6-ZmCas9 and pAtU6-2gR-ZmCas9) targeting the *AcPDS* gene were tested; only pAtU6-2gR-ZmCas9 (Figure 6c) showed positive results. On the third day post-transfection, DNA was extracted from areca protoplasts transfected with pAtU6-ZmCas9, pAtU6-2gR-ZmCas9, and from wild-type (WT) protoplasts as a control, followed by PCR amplification of the target region (Figure 6d). The amplified DNA was analyzed using the high-throughput mutation tracking platform Hi-TOM, which revealed that only the pAtU6-2gR-ZmCas9 vector successfully induced mutations at the *AcPDS* target site in areca protoplasts, with a deletion rate of 2.82% (Figure 6e). Sequencing of the edited region further confirmed the presence of insertions/deletions (indels) that were predominantly small deletions. In contrast, no detectable mutations were observed in the target region of protoplasts transfected with pAtU6-ZmCas9.

## 4. Discussion

Protoplast transient expression systems are widely utilized in plant functional genomics and gene editing to study gene function, protein subcellular localization, and intermolecular protein interactions [30,32]. Transient gene expression based on plant protoplasts is a quick and efficient method, which can compensate for the limitations of other gene function research methods for species that have difficulties with stable genetic transformation [33]. Despite their importance, an efficient protoplast transient expression system for areca palm has not yet been established. Wang et al. [17] developed a method for isolating and transforming protoplasts from fresh young leaf tissue of areca palm, but its application is limited by the extensive time required and the need for specific planting materials. Successful protoplast transient expression systems depend on a high yield of viable protoplasts from healthy plant tissues [34]. The choice of plant materials is, therefore, crucial, as it directly influences protoplast yield and viability, which are essential for downstream applications such as transfection and fusion. Furthermore, tissue source has been shown to impact the efficiency and quality of protoplast isolation [35]. Oil palm and coconut commonly utilize friable calli, liquid endosperm, or young leaves as source materials for protoplast isolation and transformation [16,36]. In this study in areca palm, we investigated four different plant tissue types (the callus, white leaves, light green leaves, and dark green leaves) to identify the most suitable source for protoplast isolation. Among these, white leaves yielded the highest number of protoplasts, highlighting their potential as an optimal tissue source. The unique physiological characteristics of white leaves, such as their softer cell walls and reduced chlorophyll content, likely facilitate the enzymatic digestion and cell separation during the isolation process. These features may have contributed to their high protoplast yield and viability. Protoplast viability was consistently high across the three leaf sections, including white, light green, and dark green leaves, indicating their overall suitability for downstream applications such as transient expression and gene editing. However, the protoplasts isolated from callus tissue exhibited lower viability, likely due to the heterogeneous nature of callus cells, which often include a mix of dedifferentiated and dead cells, as well as the potential effects of prolonged culture conditions on cell integrity. In contrast, the high yield and viability of protoplasts from white leaves made them the optimal tissue source for subsequent transformation experiments. These results highlight the need for optimization or alternative strategies to improve the use of the callus as a protoplast source for transformation. The concentration of CaCl_2_ plays a critical role in influencing transformation efficiency. Previous studies [37,38] have highlighted the impact of calcium concentration on transfection efficiency. In our study, we tested varying concentrations of CaCl_2_ (100, 200, 300, 400, and 500 mM) in PEG solutions and observed that transformation efficiency varied across the different concentrations. The highest transformation efficiency was achieved at 400 mM CaCl_2_ highlighting the critical role of calcium ions in enhancing transformation efficiency.

The amount of DNA also significantly influences stable transformation frequency, which increases proportionally with the concentration of plasmid DNA. Our findings demonstrate that higher stable transformation rates in areca protoplasts can be achieved using high-quality plasmid DNA (≥30 μg). This trend aligns with previous observations in hooker plants [39] and poplar trees [40], where the amount of plasmid DNA used for protoplast transformation affected transformation efficiencies. Transient expression is widely utilized in fluorescence-based detection, particularly for analyzing the subcellular localization of target proteins fused with fluorescent tags [41]. In our study, the observed fluorescence patterns provide crucial insights into the subcellular localization of the introduced vectors and the efficiency of transient transformation in areca protoplasts. The ubiquitous fluorescence from the p35S-GFP vector throughout the cell membrane, cytoplasm, and nucleus confirms its successful expression and supports its role as a reliable marker for general cellular activity. In contrast, the nuclear-specific fluorescence of pNLS-mNeon and pNLS-mCherry demonstrates the effectiveness of the nuclear localization signals (NLS) in directing these proteins exclusively to the nucleus. This highlights the precision of the NLS system in achieving targeted subcellular localization. The strong colocalization of GFP and mCherry fluorescence signals in the nucleus not only confirms the functionality of the NLS tag but also underscores the compatibility of these constructs for dual-marker systems. This finding is particularly valuable for studies requiring precise subcellular protein tracking or interaction analysis in transiently transformed protoplasts.

Gene editing of protoplasts has been widely used to compare the editing efficiency of CRISPR vectors, such as in rubber and citrus plants [27,38]. The structure of the sgRNA plays a crucial role in determining the efficiency and specificity of CRISPR-Cas9-mediated gene editing [42]. Employing optimized separation and transfection protocols, we evaluated the efficacy of two CRISPR vectors, pAtU6-ZmCas9 and pAtU6-2gR-ZmCas9. While both vectors were successfully transfected, only pAtU6-2gR-ZmCas9 induced detectable mutations at the target site, with a deletion rate of 2.82%. This highlights the importance of guide RNA (gRNA) design and its critical role in determining editing efficiency. As this study focused on developing a transient protoplast-based gene-editing system, a detailed analysis of off-target effects was not conducted. The sgRNAs were carefully designed using CRISPR-P 2.0 to minimize potential off-target sites. Future research involving stable areca palm transformation will address off-target analysis to ensure genome integrity.

The regeneration of protoplasts into viable plants is a critical step for advancing plant breeding. The author of [43] developed an efficient protocol for the preparation of oil palm protoplasts and successfully regenerated healthy and fertile oil palm plants. Similarly, the author of [44] reported the regeneration of calli from date palm protoplasts. However, to date, there are no reports on the regeneration of plants from areca palm protoplasts. This remains an active area of ongoing research in our work.

## 5. Conclusions

In this study, we optimized protoplast isolation techniques for different areca palm tissues (callus, white leaves, light green leaves, and dark green leaves) and found white leaves are more suitable for protoplast isolation. We achieved a transformation efficiency of 11.85% using an optimized PEG-mediated transformation protocol and a gene editing efficiency of 2.82% targeting the *AcPDS* gene via the CRISPR/Cas9 system. These results establish a valuable foundation for protoplast-based research in the areca palm and will contribute to gene function validation and genetic improvement in the areca palm and other tropical palm crops.

## Figures and Tables

**Figure 1 plants-14-00832-f001:**
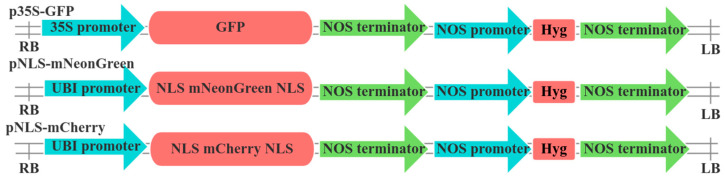
Constructs used in the current study. Structure of p35s-GFP, pNLS-mNeon, and pNLS-mCherry constructs. GFP, green fluorescent protein; NOS, Nopaline synthase; Hyg, Hygromycin; UBI (accession number: S94464), ubiquitin; NLS, nuclear localization signal; RB, Right Border; LB, Left Border.

**Figure 2 plants-14-00832-f002:**
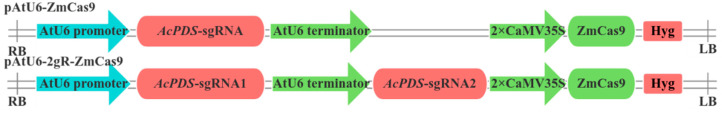
Construction of vectors. Structure of pAtU6-ZmCas9 and pAtU6-2gR-ZmCas9 constructs.

**Figure 3 plants-14-00832-f003:**
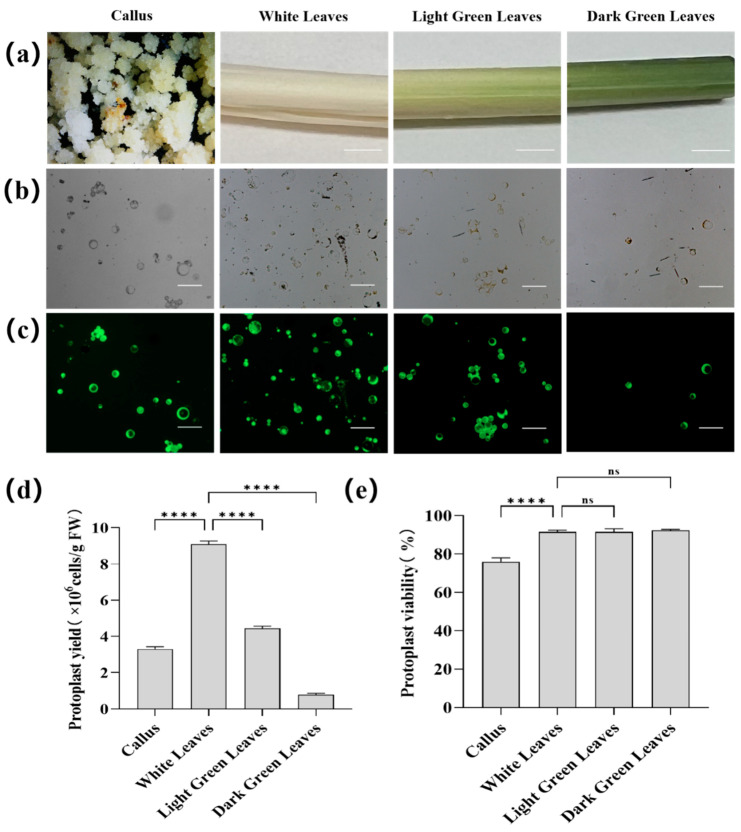
Isolation of protoplasts from areca palm. (**a**) Plant materials (callus, white leaves, light green leaves, and dark green leaves) from an areca palm plant for protoplast isolation. Scale bar = 1 cm. (**b**) Isolated protoplasts after 5 h of incubation with enzyme solution. Scale bar = 50 µm. (**c**) Viability of protoplasts observed under a fluorescence microscope. Sale Bar = 50 µm. (**d**) Yield of protoplasts isolated from the callus, white leaves, light green leaves, and dark green leaves. The number of round and intact protoplasts was counted under a light microscope. The error bar represents the mean ± standard deviation of three replicates. Significant differences between samples are indicated by asterisks (**** *p* < 0.0001), ns (not significant). (**e**) Viability of protoplasts isolated from the callus, white leaves, light green leaves, and dark green leaves. The error bar represents the mean ± standard deviation of three replicates. Significant differences between samples are indicated by asterisks (**** *p* < 0.0001), ns (not significant).

**Figure 4 plants-14-00832-f004:**
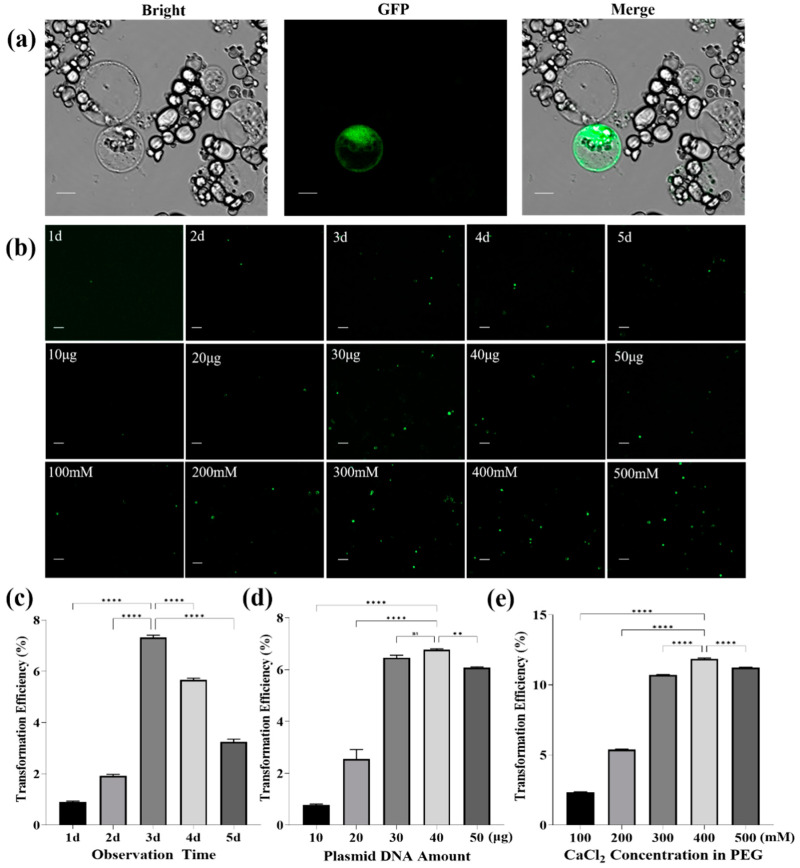
Factors influencing the transformation efficiency of areca protoplasts. (**a**) Observation of GFP expression in protoplasts after 16 h of incubation. Scale bar = 10 µm. (**b**,**c**) The effect of observation time on transformation efficiency. (**b**,**d**) The effect of plasmid DNA amount on transformation efficiency. (**b**,**e**) The effect of CaCl_2_ concentrations in PEG on transformation efficiency. Scale bar = 100 µm. Error bars represent the mean ± standard deviation of three replicates. Significant differences between samples are indicated by asterisks (** *p* < 0.01, **** *p* < 0.0001), ns (not significant).

**Figure 5 plants-14-00832-f005:**
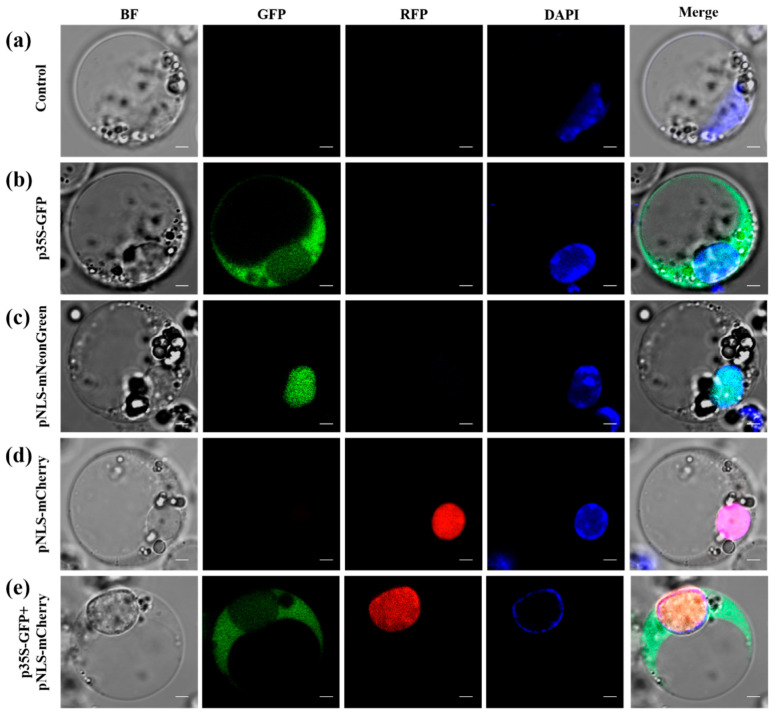
Subcellular localization in the areca protoplast. (**a**) Control. (**b**) Nuclear localization of p35s-GFP. (**c**) Nuclear localization of pNLS-mNeon. (**d**) Nuclear localization of pNLS-mCherry. (**e**) Colocalization of p35s-GFP and pNLS-mCherry vectors. Scale bar = 2 µm.

**Figure 6 plants-14-00832-f006:**
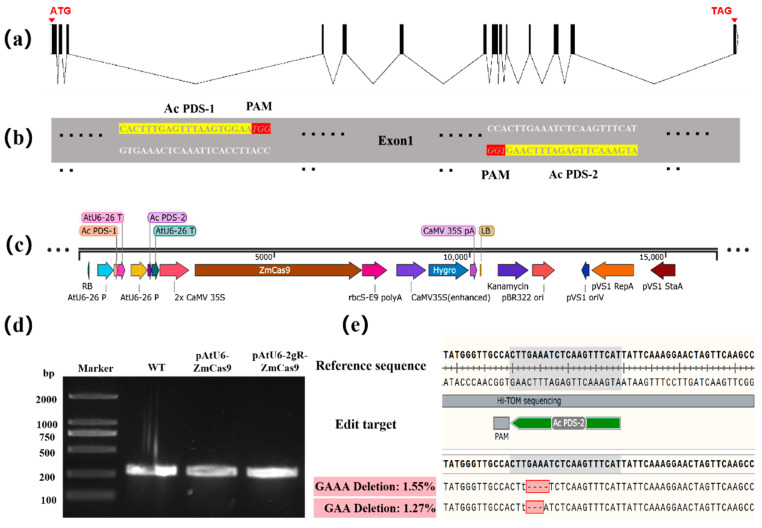
CRISPR/Cas9-based gene editing system in areca protoplasts. (**a**) *AcPDS* gene structure. (**b**) gRNAs design and in vitro cleavage assay of multiple gRNAs targeting *AcPDS* gene loci. (**c**) The schematic illustrations of the location of pAtU6-2gR-ZmCas9 target sequences. (**d**) PCR amplification of target region for areca protoplasts transformed with WT, pAtU6-ZmCas9, and pAtU6-2gR-ZmCas9. (**e**) Detection of indels in the pAtU6-2gR-ZmCas9 transfected protoplasts.

**Table 1 plants-14-00832-t001:** List of primers used for expression vector construction.

Plasmid	Primer	Sequences (5′-3′)
pNLS-mNeon	p6-UBI F	gagctcggtacccggggatctgcagcgtgacccggtcgtg
	UBI-NLS R	ccatggtggcctgcagaagtaacaccaaacaacagg
	UBI-NLS F	acttctgcaggccaccatggactataaggaccac
	NLS-mNeon R	tgctcaccatggctgctgggactccgtggata
	NLS-mNeon F	cccagcagccatggtgagcaagggcgaggagg
	mNeon-NLS R	ccggccttttcttgtacagctcgtccatgcc
	mNeon-NLS F	gctgtacaagaaaaggccggcggccacgaa
	NLS-p6 R	acgacggccagtgccaagctactccccatgggaattcgta
pNLS-mCherry	p6-UBI F	acgacggccagtgccaagcttgcagcgtgacccggtcgtg
	UBI-NLS R	ccatggtggcctgcagaagtaacaccaaac
	UBI-NLS F	acttctgcaggccaccatggactataagga
	NLS-mCherry R	tgctcaccatggctgctgggactccgtgga
	NLS-mCherry F	cccagcagccatggtgagcaagggcgagga
	mCherry-NLS R	ccggccttttcttatacagctcgtccatgc
	mCherry-NLS F	gctgtataagaaaaggccggcggccacgaa
	NLS-p6 R	gagctcggtacccggggatccgatcgggaggatcttactt

**Table 2 plants-14-00832-t002:** Single guide RNA (sgRNAs) sequence for gene editing.

Plasmid	sgRNA	Sequences (5′-3′)
pAtU6-ZmCas9	*AcPDS*-sgRNA	caaagtggagatggatccaa
pAtU6-2gR-ZmCas9	*AcPDS*-sgRNA1	cactttgagtttaagtggaa
	*AcPDS*-sgRNA2	atgaaacttgagatttcaag

## Data Availability

The original contributions presented in the study are included in the article; further inquiries can be directed to the corresponding author.

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
