# Peer review of "Development of Protoplast-Based Gene Editing System for Areca Palm"

_plants, 2025, doi:10.3390/plants14060832_

Round 1
Reviewer 1 Report
Comments and Suggestions for Authors
1. In the Graphical abstract, the bright - field protoplast image does not correspond to the FDA fluorescence image. It is recommended to replace the photo.
2. For Figure 3a, it is recommended to add the corresponding bright - field image of FDA.
3. For Figure 4b, it is recommended to supplement the corresponding bright - field photo to clearly show the transformation efficiency. There is no significant difference in the number of fluorescent protoplasts in the images of the 3rd, 4th, and 5th days, which is not quite consistent with the result in Figure (c). It is suggested to replace the pictures.
Author Response
Thank you very much for taking the time to review this manuscript. Please find the attached document containing our responses and revisions.

Reviewer 2 Report
Comments and Suggestions for Authors
The manuscript submitted by Nie et al. details the development of a transient expression assay system in areca palm protoplasts. The authors demonstrated transient expression of the marker GFP gene, nuclear localization of other fluorescent marker proteins, and CRISPR/Cas9-mediated endogenous PDS gene mutagenesis. Although the authors have not attempted to regenerate areca palm plants from the generated protoplasts, the topics described are of importance to the relevant community. However, some issues have been identified that need to be addressed and corrected.
The subtitle "optimized protocol for protoplast isolation" (line 214) appears to be an inaccurate representation of the experimental protocol. This is because the authors did not conduct their research with the objective of optimizing protoplast isolation experiments. The hypothesis that white leaves are the optimal substrate for protoplast isolation must be discussed.
It is imperative that the rationale behind the selection of the four sample types for protoplast isolation be elucidated in the manuscript.
A detailed description of the statistical method employed by the authors to analyze the significance of Figures 3d, e, 4c-e is necessary. It appears that the authors employed Student's t-test for the analysis. However, it is imperative to note that, in the context of analyzing significant differences among multiple samples, the ANOVA with Tukey's multiple comparisons test is the appropriate statistical approach, as opposed to the Student t-test.
It appears that the number of designed sgRNAs is not consistent. The text and Figure 6b indicate the presence of two sgRNAs; however, Table 2 describes three distinct sgRNAs.
What is the UBQ Promoter? A more detailed explanation is needed. Why are there two NLSs in mNeonGreen and mCherry (Table 1 also)? Why does Hyg not have a promoter and terminator? (Fig. 2 also)
The CRISPR/Cas9-mediated mutagenesis experiment appears to be inadequately designed. The authors must provide a thorough description of the ZmCas9 and the rationale behind its utilization, along with the rationale for the employment of the AtU6 promoter. The experimental design exhibits notable inconsistencies. The ZmCas9 is supposed to be the maize codon-optimized Cas9, which would be reasonable because palm is a monocot plant. The utilization of 35s and AtU6 promoters in this context warrants further examination, particularly in light of the recognized diminished transcriptional activities of these promoters in monocot plants. It is therefore recommended that the Cas9 system, which has been well-established in rice or other monocot plants, be employed in future studies to ensure optimal results.
The significance of the length of the sgRNA remains unclear (line 372). According to the most recent literature on the subject, a 20-base pair protospacer sequence is typically employed for the design of sgRNAs.
An analysis of the off-target effect of Cas9 is necessary.
A query is raised as to whether there exists any documented evidence that validates the efficacy of Cas9 constructs in achieving successful transfection (line 376).
Minor comments
It should be written in a unified way, such as mNeonGreen and mNeon.
Author Response
Thank you very much for taking the time to review the manuscript. Please find the attached document containing our responses and revisions.

Reviewer 3 Report
Comments and Suggestions for Authors
some minor grammatical changes suggested on the attached.
In order for this invention to have utility in plant breeding, it is necessary to be able to regenerate palm plants from protoplasts. I didn't see this topic mentioned. Please review the progress in regenerating protoplasts from palm plants in the discussion.

Author Response
Thank you very much for taking the time to review our manuscript. Please find the attached document containing our responses and revisions.

Round 2
Reviewer 2 Report
Comments and Suggestions for Authors
The authors offered a response to my comments in the cover letter; however, the majority of these comments have not been reflected in the manuscript. Moreover, I still have some concerns that need to be clarified.
The most worrisome aspect of this manuscript is that the authors have yet to provide a thorough description of the ZmCas9 and the rationale behind its utilization. In addition, the rationale behind the employment of the AtU6 promoter remains unclear. Furthermore, the design of the CRISPR/Cas9-mediated mutagenesis experiment has been inadequately formulated. It is imperative that the manuscript include a clear explanation of the experimental design.
The rationale behind the authors' inclusion of the length of sgRNA in the manuscript, as evidenced by lines 79 and 391, remains ambiguous. Was an examination conducted on the optimized length of protospacer sequence of CRISPR/Cas9 for arca palm? In the event that such an examination was conducted, the results should be presented.
It is imperative to ascertain the potential for off-target sites to be predicted by the CRISPR-P 2.0 tool. As the authors indicated in the manuscript, this is the inaugural report of CRISPR/Cas9-mediated mutagenesis in areca palm. While the present study did not involve the regeneration of plants from protoplasts, the potential for off-target effects in such processes warrants further examination. Such findings are likely to be of significant value to the scientific community and to subsequent research endeavors.
It should be noted that the incubation temperature of the transformed areca palm protoplast.
Point 5: I would like to express my sincere apologies for the ambiguity in my previous comment. The aforementioned comment pertained to the source of the UBQ (or UBI) promoter and the accession number of the promoter. It is imperative that this information be clarified in the manuscript.
The manuscript fails to clarify the number of sgRNAs that were designed. According to the authors' response, three distinct sgRNAs were designed, and one was utilized for the single sgRNA construct, while the other two were used for the double sgRNA construct. It is imperative that these details be explicitly delineated in the manuscript to ensure its clarity and integrity.
As previously mentioned, the following comment pertains to the subject. The authors designed three distinct sgRNAs, yet only one of them demonstrated a positive result, exhibiting an efficiency of less than 3%. The number of positive results is inadequate. To further explore the potential of the CRISPR/Cas9 system in areca palm, it is recommended that additional experiments be conducted, particularly focusing on the utilization of additional positive sgRNA experiments. Secondly, the mutagenesis efficiency observed in areca palm protoplasts requires further elucidation. The transformation efficiency observed in this study is presumed to be sufficiently high for the subsequent execution of additional experiments. Consequently, it is hypothesized that the low mutagenesis efficiency can be attributed to two potential factors: the design of sgRNA, and the design of the CRISPR/Cas9 experiment itself. It is therefore strongly recommended that the authors perform CRISPR/Cas9 experiments using the well-established Cas9 system in rice or other monocot plants to ensure optimal results.
Comments on the Quality of English Language
For my opinion, the quality of English on this paper is acceptable, the text is partly fluent but not always.
Author Response
Thank you very much for taking the time to review this manuscript. Please see the attached files containing detailed responses and the corresponding revisions in track changes.

Round 3
Reviewer 2 Report
Comments and Suggestions for Authors
I apologize that my comments were not clear.
The length of sgRNA
As the authors note in their comments, the 20 bp protospacer sequence is widely used for Cas9-mediated mutagenesis in a wide range of organisms. The authors also applied the 20 bp protospacer sequence in this study. Therefore, it is recognized and common knowledge that the 20 bp protospacer sequence is an optimized length. It is unclear to me why the authors mentioned the sgRNA length in this manuscript. It is thought that there is no need to mention it in the present study (lines 81 and 369).
Cas9-mediated mutagenesis experiments
As I mentioned in my previous comment, I do not think that Cas9-based mutagenesis experiments are properly designed. Usually, in monocotyledons, a ZmUBI promoter (or 35s promoter) driven Cas9 (ZmCas9 or regular Cas9, not a version optimized for dicotyledonous plants codons) is applied together with OsU6-sgRNA. Therefore, in this study, the use of the AtU6 promoter instead of the OsU6 promoter, which is widely used in monocot plants, requires clarification to the readers. It does not encourage the authors to further study using the AcU6 promoter and the areca palm codon optimized Cas9. My point is that a clear explanation of the experimental design as to why the AtU6 promoter was used on the monocot areca palm should be clearly included in the manuscript for the benefit of the readers.
Author Response
Thank you very much for taking the time to review this manuscript. Please find our detailed responses in the attached file, along with the corresponding revisions highlighted in track changes in the Revised Manuscript.

Round 4
Reviewer 2 Report
Comments and Suggestions for Authors
Most of my concerns were addressed and improved in the revised manuscript.